# Online Learning with a Hint

**Ofer Dekel**
Microsoft Research
oferd@microsoft.com

**Arthur Flajolet**
Operations Research Center
Massachusetts Institute of Technology
flajolet@mit.edu

**Nika Haghtalab**
Computer Science Department
Carnegie Mellon University
nika@cmu.edu

**Patrick Jaillet**
EECS, LIDS, ORC
Massachusetts Institute of Technology
jaillet@mit.edu

## Abstract

We study a variant of online linear optimization where the player receives a hint about the loss function at the beginning of each round. The hint is given in the form of a vector that is weakly correlated with the loss vector on that round. We show that the player can benefit from such a hint if the set of feasible actions is sufficiently round. Specifically, if the set is strongly convex, the hint can be used to guarantee a regret of $O(\log(T))$, and if the set is $q$-uniformly convex for $q \in (2, 3)$, the hint can be used to guarantee a regret of $o(\sqrt{T})$. In contrast, we establish $\Omega(\sqrt{T})$ lower bounds on regret when the set of feasible actions is a polyhedron.

## 1  Introduction

Online linear optimization is a canonical problem in online learning. In this setting, a player attempts to minimize an online adversarial sequence of loss functions while incurring a small *regret*, compared to the best offline solution. Many online algorithms exist that are designed to have a regret of $O(\sqrt{T})$ in the worst case and it has been known that one cannot avoid a regret of $\Omega(\sqrt{T})$ in the worst case. While this worst-case perspective on online linear optimization has lead to elegant algorithms and deep connections to other fields, such as boosting [9, 10] and game theory [4, 2], it can be overly pessimistic. In particular, it does not account for the fact that the player may have side-information that allows him to anticipate the upcoming loss functions and evade the $\Omega(\sqrt{T})$ regret. In this work, we go beyond this worst case analysis and consider online linear optimization when additional information in the form of a function that is correlated with the loss is presented to the player.

More formally, online convex optimization [24, 11] is a $T$-round repeated game between a player and an adversary. On each round, the player chooses an action $x_t$ from a convex set of feasible actions $\mathcal{K} \subseteq \mathbb{R}^d$ and the adversary chooses a convex bounded loss function $f_t$. Both choices are revealed and the player incurs a loss of $f_t(x_t)$. The player then uses its knowledge of $f_t$ to adjust its strategy for the subsequent rounds. The player's goal is to accumulate a small loss compared to the best fixed action in hindsight. This value is called *regret* and is a measure of success of the player's algorithm.

When the adversary is restricted to Lipschitz loss functions, several algorithms are known to guarantee $O(\sqrt{T})$ regret [24, 16, 11]. If we further restrict the adversary to strongly convex loss functions, the regret bound improves to $O(\log(T))$ [14]. However, when the loss functions are linear, no online algorithm can have a regret of $o(\sqrt{T})$ [5]. In this sense, linear loss functions are the most difficult convex loss functions to handle [24].

In this paper, we focus on the case where the adversary is restricted to linear Lipschitz loss functions. More specifically, we assume that the loss function $f_t(x)$ takes the form $c_t^\mathsf{T} x$, where $c_t$ is a bounded loss vector in $\mathcal{C} \subseteq \mathbb{R}^d$. We further assume that the player receives a *hint* before choosing the action on each round. The hint in our setting is a vector that is guaranteed to be weakly correlated with the loss vector. Namely, at the beginning of round $t$, the player observes a unit-length vector $v_t \in \mathbb{R}^d$ such that $v_t^\mathsf{T} c_t \geq \alpha \|c_t\|_2$, and where $\alpha$ is a small positive constant. So long as this requirement is met, the hint could be chosen maliciously, possibly by an adversary who knows how the player's algorithm uses the hint. Our goal is to develop a player strategy that takes these hints into account, and to understand when hints of this type make the problem provably easier and lead to smaller regret.

We show that the player's ability to benefit from the hints depends on the geometry of the player's action set $\mathcal{K}$. Specifically, we characterize the roundness of the set $\mathcal{K}$ using the notion of $(C, q)$-uniform convexity for convex sets. In Section 3, we show that if $\mathcal{K}$ is a $(C, 2)$-uniformly convex set (or in other words, if $\mathcal{K}$ is a $C$-strongly convex set), then we can use the hint to design a player strategy that improves the regret guarantee to $O\big((C\alpha)^{-1} \log(T)\big)$, where our $O(\cdot)$ notation hides a polynomial dependence on the dimension $d$ and other constants. Furthermore, as we show in Section 4, if $\mathcal{K}$ is a $(C, q)$-uniformly convex set for $q \in (2, 3)$, we can use the hint to improve the regret to $O\left((C\alpha)^{\frac{1}{1-q}} T^{\frac{2-q}{1-q}}\right)$, when the hint belongs to a small set of possible hints at every step.

In Section 5, we prove lower bounds on the regret of any online algorithm in this model. We first show that when $\mathcal{K}$ is a polyhedron, such as a $L_1$ ball, even a stronger form of hint cannot guarantee a regret of $o(\sqrt{T})$. Next, we prove a lower bound of $\Omega(\log(T))$ regret when $\mathcal{K}$ is strongly convex.

## 1.1 Comparison with Other Notions of Hints

The notion of hint that we introduce in this work generalizes some of the notions of predictability on online learning. Hazan and Megiddo [13] considered as an example a setting where the player knows the first coordinate of the loss vector at all rounds, and showed that when $|c_{t1}| \geq \alpha$ and when the set of feasible actions is the Euclidean ball, one can achieve a regret of $O(1/\alpha \cdot \log(T))$. Our work directly improves over this result, as in our setting a hint $v_t = \pm e_1$ also achieves $O(1/\alpha \cdot \log(T))$ regret, but we can deal with hints in different directions at different rounds and we allow for general uniformly convex action sets. Rakhlin and Sridharan [20] considered online learning with predictable sequences, with a notion of predictability that is concerned with the gradient of the convex loss functions. They show that if the player receives a hint $M_t$ at round $t$, then the regret of the algorithm is at most $O(\sqrt{\sum_{t=1}^T \|\nabla f_t(x_t) - M_t\|_*^2})$.

In the case of linear loss functions, this implies that having an estimate vector $c_t'$ of the loss vector within distance $\sigma$ of the true loss vector $c_t$ results in an improved regret bound of $O(\sigma\sqrt{T})$. In contrast, we consider a notion of hint that pertains to the *direction of the loss vector* rather than its location. Our work shows that merely knowing whether the loss vector positively or negatively correlates with another vector is sufficient to achieve improved regret bound, when the set is uniformly convex. That is, rather than having access to an approximate value of $c_t$, we only need to have access to a halfspace that classifies $c_t$ correctly with a margin. This notion of hint is weaker that the notion of hint in the work of Rakhlin and Sridharan [20] when the approximation error satisfies $\|c_t - c_t'\|_2 \leq \sigma \cdot \|c_t\|_2$ for $\sigma \in [0, 1)$. In this case one can use $c_t'/\|c_t'\|_2$ as the direction of the hint in our setting and achieve a regret of $O(\frac{1}{1-\sigma} \log T)$ when the set is strongly convex. This shows that when the set of feasible actions is strongly convex, a directional hint can improve the regret bound beyond what has been known to be achievable by an approximation hint. However, we note that our results require the hints to be always valid, whereas the algorithm of Rakhlin and Sridharan [19] can adapt to the quality of the hints.

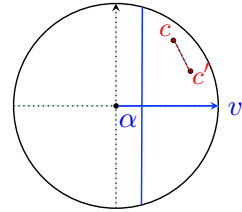

Figure 1: Comparison between notions of hint.

We discuss these works and other related works, such as [15], in more details in Appendix A.

## 2 Preliminaries

We begin with a more formal definition of online linear optimization (without hints). Let $\mathcal{A}$ denote the player's algorithm for choosing its actions. On round $t$ the player uses $\mathcal{A}$ and all of the information

it has observed so far to choose an action $x_t$ in a convex compact set $\mathcal{K} \subseteq \mathbb{R}^d$. Subsequently, the adversary chooses a loss vector $c_t$ in a compact set $\mathcal{C} \subseteq \mathbb{R}^d$. The player and the adversary reveal their actions and the player incurs the loss $c_t^\intercal x_t$. The player's regret is defined as

$$R(\mathcal{A}, c_{1:T}) \;=\; \sum_{t=1}^{T} c_t^\intercal x_t - \min_{x \in \mathcal{K}} \sum_{t=1}^{T} c_t^\intercal x.$$

In online linear optimization *with hints*, the player observes $v_t \in \mathbb{R}^d$ with $\|v_t\|_2 = 1$, before choosing $x_t$, with the guarantee that $v_t^\intercal c_t \geq \alpha \|c_t\|_2$, for some $\alpha > 0$.

We use *uniform convexity* to characterize the degree of convexity of the player's action set $\mathcal{K}$. Informally, uniform convexity requires that the convex combination of any two points $x$ and $y$ on the boundary of $\mathcal{K}$ be sufficiently far from the boundary. A formal definition is given below.

**Definition 2.1** (Pisier [18]). *Let $\mathcal{K}$ be a convex set that is symmetric around the origin. $\mathcal{K}$ and the Banach space defined by $\mathcal{K}$ are said to be* uniformly convex *if for any $0 < \epsilon < 2$ there exists a $\delta > 0$ such that for any pair of points $x, y \in \mathcal{K}$ with $\|x\|_\mathcal{K} \leq 1, \|y\|_\mathcal{K} \leq 1, \|x - y\|_\mathcal{K} \geq \epsilon$, we have $\left\| \frac{x+y}{2} \right\|_\mathcal{K} \leq 1 - \delta$. The* modulus of uniform-convexity $\delta_\mathcal{K}(\epsilon)$ *is the best possible $\delta$ for that $\epsilon$, i.e.,*

$$\delta_\mathcal{K}(\epsilon) = \inf \left\{ 1 - \left\| \frac{x+y}{2} \right\|_\mathcal{K} : \|x\|_\mathcal{K} \leq 1, \|y\|_\mathcal{K} \leq 1, \|x - y\|_\mathcal{K} \geq \epsilon \right\}.$$

*For brevity, we say that $\mathcal{K}$ is $(C, q)$-uniformly convex when $\delta_\mathcal{K}(\epsilon) = C\epsilon^q$ and we omit $C$ when it is clear from the context.*

Examples of uniformly convex sets include $L_p$ balls for any $1 < p < \infty$ with modulus of convexity $\delta_{L_p}(\epsilon) = C_p \epsilon^p$ for $p \geq 2$ and a constant $C_p$ and $\delta_{L_p}(\epsilon) = (p-1)\epsilon^2$ for $1 < p \leq 2$. On the other hand, $L_1$ and $L_\infty$ units balls are not uniformly convex. When $\delta_\mathcal{K}(\epsilon) \in \Theta(\epsilon^2)$, we say that $\mathcal{K}$ is *strongly convex*.

Another notion of convexity we use in this work is called *exp-concavity*. A function $f : \mathcal{K} \to \mathbb{R}$ is exp-concave with parameter $\beta > 0$, if $\exp(-\beta f(x))$ is a concave function of $x \in \mathcal{K}$. This is a weaker requirement than strong convexity when the gradient of $f$ is uniformly bounded [14]. The next proposition shows that we can obtain regret bounds of order $\Theta(\log(T))$ in online convex optimization when the loss functions are exp-concave with parameter $\beta$.

**Proposition 2.2** ([14]). *Consider online convex optimization on a sequence of loss functions $f_1, \ldots, f_T$ over a feasible set $\mathcal{K} \subseteq \mathbb{R}^d$, such that all $t$, $f_t : \mathcal{K} \to \mathbb{R}$ is exp-concave with parameter $\beta > 0$. There is an algorithm, with runtime polynomial in d, which we call $\mathcal{A}_{\mathrm{EXP}}$, with a regret that is at most $\frac{d}{\beta}(1 + \log(T+1))$.*

Throughout this work, we draw intuition from basic orthogonal geometry. Given any vector $x$ and a hint $v$, we define $x^{\|v} = (x^\intercal v)v$ and $x^{\perp v} = x - (x^\intercal v)v$, as the parallel and the orthogonal components of $x$ with respect to $v$. When the hint $v$ is clear from the context we simply use $x^\|$ and $x^\perp$ to denote these vectors.

Naturally, our regret bounds involve a number of geometric parameters. Since the set of actions of the adversary $\mathcal{C}$ is compact, we can find $G \geq 0$ such that $\|c\|_2 \leq G$ for all $c \in \mathcal{C}$. When $\mathcal{K}$ is uniformly convex, we denote $\mathcal{K} = \{w \in \mathbb{R}^d \mid \|w\|_\mathcal{K} \leq 1\}$. In this case, since all norms are equivalent in finite dimension, there exist $R > 0$ and $r > 0$ such that $B_r \subseteq \mathcal{K} \subseteq B_R$, where $B_r$ (resp. $B_R$) denote the $L_2$ unit ball centered at 0 with radius $r$ (resp. $R$). This implies that $\frac{1}{R}\|\cdot\|_2 \leq \|\cdot\|_\mathcal{K} \leq \frac{1}{r}\|\cdot\|_2$.

## 3 Improved Regret Bounds for Strongly Convex $\mathcal{K}$

At first sight, it is not immediately clear how one should use the hint. Since $v_t$ is guaranteed to satisfy $c_t^\intercal v_t \geq \alpha \|c_t\|_2$, moving the action $x$ in the direction $-v_t$ always decreases the loss. One could hope to get the most benefit out of the hint by choosing $x_t$ to be the extremal point in $\mathcal{K}$ in the direction $-v_t$. However, this naïve strategy could lead to a linear regret in the worst case. For example, say that $c_t = (1, \frac{1}{2})$ and $v_t = (0, 1)$ for all $t$ and let $\mathcal{K}$ be the Euclidean unit ball. Choosing $x_t = -v_t$ would incur a loss of $-\frac{T}{2}$, while the best fixed action in hindsight, the point $(\frac{-2}{\sqrt{5}}, \frac{-1}{\sqrt{5}})$, would incur a loss of $\frac{-\sqrt{5}}{2}T$. The player's regret would therefore be $\frac{\sqrt{5}-1}{2}T$.

Intuitively, the flaw of this naïve strategy is that the hint does not give the player any information about the $(d-1)$-dimensional subspace orthogonal to $v_t$. Our solution is to use standard online learning machinery to learn how to act in this orthogonal subspace. Specifically, on round $t$, we use $v_t$ to define the following *virtual loss function*:

$$\hat{c}_t(x) = \min_{w \in \mathcal{K}} c_t^\mathsf{T} w \quad \text{s.t.} \quad w^{\perp v_t} = x^{\perp v_t} \ .$$

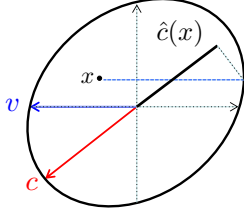

Figure 2: Virtual function $\hat{c}(\cdot)$.

In words, we consider the 1-dimensional subspace spanned by $v_t$ and its $(d-1)$-dimensional orthogonal subspace separately. For any action $x \in \mathcal{K}$, we find another point, $w \in \mathcal{K}$, that equals $x$ in the $(d-1)$-dimensional orthogonal subspace, but otherwise incurs the optimal loss. The value of the virtual loss $\hat{c}_t(x)$ is defined to be the value of the original loss function $c_t$ at $w$. The virtual loss simulates the process of moving $x$ as far as possible in the direction $-v_t$ without changing its value in any other direction (see Figure 2). This can be more formally seen by the following equation.

$$\underset{w \in \mathcal{K}: w^\perp = \hat{x}^\perp}{\arg\min} c_t^\mathsf{T} w = \underset{w \in \mathcal{K}: w^\perp = \hat{x}^\perp}{\arg\min} \left( (c_t^\perp)^\mathsf{T} \hat{x}^\perp + (c_t^\|)^\mathsf{T} w^\| \right) = \underset{w \in \mathcal{K}: w^\perp = \hat{x}^\perp}{\arg\min} v_t^\mathsf{T} w, \qquad (1)$$

where the last transition holds by the fact that $c_t^\| = \left\| c_t^\| \right\|_2 v_t$ since the hint is valid.

This provides an intuitive understanding of a measure of convexity of our virtual loss functions. When $\mathcal{K}$ is uniformly convex then the function $\hat{c}_t(\cdot)$ demonstrates convexity in the subspace orthogonal to $v_t$. To see that, note that for any $x$ and $y$ that lie in the space orthogonal to $v_t$, their mid point $\frac{x+y}{2}$ transforms to a point that is farther away in the direction of $-v_t$ than the midpoint of the transformations of $x$ and $y$. As shown in Figure 3, the modulus of uniform convexity of $\mathcal{K}$ affects the degree of convexity of $\hat{c}_t(\cdot)$. We note, however, that $\hat{c}_t(\cdot)$ is not strongly convex in all directions. In fact, $\hat{c}_t(\cdot)$ is constant in the direction of $v_t$. Nevertheless, the properties shown here allude to the fact that $\hat{c}_t(\cdot)$ demonstrates some notion of convexity. As we show in the next lemma, this notion is indeed *exp-concavity*:

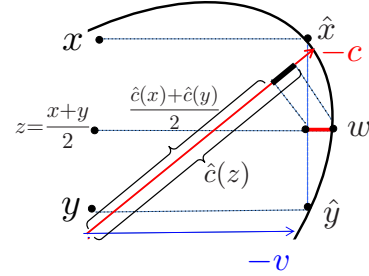

Figure 3: Uniform-convexity of the feasible set affects the convexity the virtual loss function.

**Lemma 3.1.** *If $\mathcal{K}$ is $(C, 2)$-uniformly convex, then $\hat{c}_t(\cdot)$ is $8 \frac{\alpha \cdot C \cdot r}{G \cdot R^2}$-exp-concave.*

*Proof.* Let $\gamma = 8 \frac{\alpha \cdot C \cdot r}{G \cdot R^2}$. Without loss of generality, we assume that $c_t \neq 0$, otherwise $\hat{c}_t(\cdot) = 0$ is a constant function and the proof follows immediately. Based on the above discussion, it is not hard to see that $\hat{c}_t(\cdot)$ is continuous (we prove this in more detail in the Appendix D.1. So, to prove that $\hat{c}_t(\cdot)$ is exp-concave, it is sufficient to show that

$$\exp\left( -\gamma \cdot \hat{c}_t \left( \frac{x+y}{2} \right) \right) \geq \frac{1}{2} \exp\left( -\gamma \cdot \hat{c}_t(x) \right) + \frac{1}{2} \exp\left( -\gamma \cdot \hat{c}_t(y) \right) \quad \forall (x, y) \in \mathcal{K}.$$

Consider $(x, y) \in \mathcal{K}$ and choose corresponding $(\hat{x}, \hat{y}) \in \mathcal{K}$ such that $\hat{c}_t(x) = c_t^\mathsf{T} \hat{x}$ and $\hat{c}_t(y) = c_t^\mathsf{T} \hat{y}$. Without loss of generality, we have $\|\hat{x}\|_\mathcal{K} = \|\hat{y}\|_\mathcal{K} = 1$, as we can always choose corresponding $\hat{x}, \hat{y}$ that are extreme points of $\mathcal{K}$. Since $\exp(-\gamma \hat{c}_t(\cdot))$ is decreasing in $\hat{c}_t(\cdot)$, we have

$$\exp\left( -\gamma \cdot \hat{c}_t \left( \frac{x+y}{2} \right) \right) = \max_{\substack{\|w\|_\mathcal{K} \leq 1 \\ w^{\perp v_t} = (\frac{x+y}{2})^{\perp v_t}}} \exp(-\gamma \cdot c_t^\mathsf{T} w). \qquad (2)$$

Note that $w = \frac{\hat{x}+\hat{y}}{2} - \delta_\mathcal{K}(\|\hat{x} - \hat{y}\|_\mathcal{K}) \frac{v_t}{\|v_t\|_\mathcal{K}}$ satisfies $\|w\|_\mathcal{K} \leq 1$, since $\|w\|_\mathcal{K} \leq \left\| \frac{\hat{x}+\hat{y}}{2} \right\|_\mathcal{K} + \delta_\mathcal{K}(\|\hat{x} - \hat{y}\|_\mathcal{K}) \leq 1$ (see also Figure 3). Moreover, $w^{\perp v_t} = (\frac{x+y}{2})^{\perp v_t}$. So, by using this $w$ in Equation (2), we have

$$\exp\left( -\gamma \cdot \hat{c}_t \left( \frac{x+y}{2} \right) \right) \geq \exp\left( -\frac{\gamma}{2} \cdot (c_t^\mathsf{T} \hat{x} + c_t^\mathsf{T} \hat{y}) + \gamma \cdot \frac{c_t^\mathsf{T} v_t}{\|v_t\|_\mathcal{K}} \cdot \delta_\mathcal{K}(\|\hat{x} - \hat{y}\|_\mathcal{K}) \right). \qquad (3)$$

On the other hand, since $\|v_t\|_\mathcal{K} \leq \frac{1}{r}\|v_t\|_2 = \frac{1}{r}$ and $\|\hat{x} - \hat{y}\|_\mathcal{K} \geq \frac{1}{R}\|\hat{x} - \hat{y}\|_2$, we have

$$\exp\left(\gamma \cdot \frac{c_t^\mathsf{T} v_t}{\|v_t\|_\mathcal{K}} \cdot \delta_\mathcal{K}(\|\hat{x} - \hat{y}\|_\mathcal{K})\right) \geq \exp\left(\gamma \cdot r \cdot \alpha \cdot \|c_t\|_2 \cdot C \cdot \frac{1}{R^2} \cdot \|\hat{x} - \hat{y}\|_2^2\right)$$

$$\geq \exp\left(\gamma \cdot \frac{\alpha \cdot C \cdot r}{R^2} \cdot \|c_t\|_2 \cdot \left(\frac{c_t^\mathsf{T}\hat{x}}{\|c_t\|_2} - \frac{c_t^\mathsf{T}\hat{y}}{\|c_t\|_2}\right)^2\right)$$

$$\geq \exp\left(\frac{(\gamma/2)^2 \cdot (c_t^\mathsf{T}\hat{x} - c_t^\mathsf{T}\hat{y})^2}{2}\right)$$

$$\geq \frac{1}{2} \cdot \exp\left(\frac{\gamma}{2} \cdot (c_t^\mathsf{T}\hat{x} - c_t^\mathsf{T}\hat{y})\right) + \frac{1}{2} \cdot \exp\left(\frac{\gamma}{2} \cdot (c_t^\mathsf{T}\hat{y} - c_t^\mathsf{T}\hat{x})\right),$$

where the penultimate inequality follows by the definition of $\gamma$ and the last inequality is a consequence of the inequality $\exp(z^2/2) \geq \frac{1}{2}\exp(z) + \frac{1}{2}\exp(-z), \forall z \in \mathbb{R}$. Plugging the last inequality into (3) yields

$$\exp\left(-\gamma\hat{c}_t(\frac{x+y}{2})\right) \geq \frac{1}{2}\exp\left(-\frac{\gamma}{2}(c_t^\mathsf{T}\hat{x} + c_t^\mathsf{T}\hat{y})\right) \cdot \left\{\exp\left(\frac{\gamma}{2}(c_t^\mathsf{T}\hat{x} - c_t^\mathsf{T}\hat{y})\right) + \exp\left(\frac{\gamma}{2}(c_t^\mathsf{T}\hat{y} - c_t^\mathsf{T}\hat{x})\right)\right\}$$

$$= \frac{1}{2}\exp\left(-\gamma \cdot c_t^\mathsf{T}\hat{y}\right) + \frac{1}{2}\exp\left(-\gamma \cdot c_t^\mathsf{T}\hat{x}\right)$$

$$= \frac{1}{2}\exp\left(-\gamma \cdot \hat{c}_t(y)\right) + \frac{1}{2}\exp\left(-\gamma \cdot \hat{c}_t(x)\right),$$

which concludes the proof. $\qquad\square$

Now, we use the sequence of virtual loss functions to reduce our problem to a standard online convex optimization problem (without hints). Namely, the player applies $\mathcal{A}_{\mathrm{EXP}}$ (from Proposition 2.2), which is an online convex optimization algorithm known to have $O(\log(T))$ regret with respect to exp-concave functions, to the sequence of virtual loss functions. Then our algorithm takes the action $\hat{x}_t \in \mathcal{K}$ that is prescribed by $\mathcal{A}_{\mathrm{EXP}}$ and moves it as far as possible in the direction of $-v_t$. This process is formalized in Algorithm 1.

---

**Algorithm 1** $\mathcal{A}_{\mathrm{hint}}$ FOR STRONGLY CONVEX $\mathcal{K}$

For $t = 1, \ldots, T$,

    1. Use Algorithm $\mathcal{A}_{\mathrm{EXP}}$ with the history $\hat{c}_\tau(\cdot)$ for $\tau < t$, and let $\hat{x}_t$ be the chosen action.

    2. Let $x_t = \arg\min_{w \in \mathcal{K}}(v_t^\mathsf{T}w)$ s.t. $w^{\perp v_t} = \hat{x}_t^{\perp v_t}$. Play $x_t$ and receive $c_t$ as feedback.

---

Next, we show that the regret of algorithm $\mathcal{A}_{\mathrm{EXP}}$ on the sequence of virtual loss functions is an upper bound on the regret of Algorithm 1.

**Lemma 3.2.** *For any sequence of loss functions $c_1, \ldots, c_T$, let $R(\mathcal{A}_{\mathrm{hint}}, c_{1:T})$ be the regret of algorithm $\mathcal{A}_{\mathrm{hint}}$ on the sequence $c_1, \ldots, c_T$, and $R(\mathcal{A}_{\mathrm{EXP}}, \hat{c}_{1:T})$ be the regret of algorithm $\mathcal{A}_{\mathrm{EXP}}$ on the sequence of virtual loss functions $\hat{c}_1, \ldots, \hat{c}_T$. Then, $R(\mathcal{A}_{\mathrm{hint}}, c_{1:T}) \leq R(\mathcal{A}_{\mathrm{EXP}}, \hat{c}_{1:T})$.*

*Proof.* Equation (1) provides an equivalent definition $x_t = \arg\min_{w \in \mathcal{K}}(c_t^\mathsf{T}w)$ s.t. $w^{\perp v_t} = \hat{x}_t^{\perp v_t}$. Using this, we show that the loss of algorithm $\mathcal{A}_{\mathrm{hint}}$ on the sequence $c_{1:T}$ is the same as the loss of algorithm $\mathcal{A}_{\mathrm{EXP}}$ on the sequence $\hat{c}_{1:T}$.

$$\sum_{t=1}^T \hat{c}_t(\hat{x}_t) = \sum_{t=1}^T \min_{w \in \mathcal{K}: w^\perp = \hat{x}_t^\perp} c_t^\mathsf{T}w = \sum_{t=1}^T c_t^\mathsf{T}(\arg\min_{w \in \mathcal{K}: w^\perp = \hat{x}_t^\perp} c_t^\mathsf{T}w) = \sum_{t=1}^T c_t^\mathsf{T}x_t.$$

Next, we show that the offline optimal on the sequence $\hat{c}_{1:T}$ is more competitive that the offline optimal on the sequence $c_{1:T}$. First note that for any $x$ and $t$, $\hat{c}_t(x) = \min_{w \in \mathcal{K}: w^\perp = x^\perp} c_t^\mathsf{T}w \leq c_t^\mathsf{T}x$. Therefore, $\min_{x \in \mathcal{K}} \sum_{t=1}^T \hat{c}_t(x) \leq \min_{x \in \mathcal{K}} \sum_{t=1}^T c_t^\mathsf{T}x$. The proof concludes by

$$R(\mathcal{A}_{\mathrm{hint}}, c_{1:T}) = \sum_{t=1}^T c_t^\mathsf{T}x_t - \min_{x \in \mathcal{K}} \sum_{t=1}^T c_t^\mathsf{T}x \leq \sum_{t=1}^T \hat{c}_t(\hat{x}_t) - \min_{x \in \mathcal{K}} \sum_{t=1}^T \hat{c}_t(x) = R(\mathcal{A}_{\mathrm{EXP}}, \hat{c}_{1:T}).$$

$\qquad\square$

Our main result follows from the application of Lemmas 3.1 and 3.2.

**Theorem 3.3.** *Suppose that $\mathcal{K} \subseteq \mathbb{R}^d$ is a $(C, 2)$-uniformly convex set that is symmetric around the origin, and $B_r \subseteq \mathcal{K} \subseteq B_R$ for some $r$ and $R$. Consider online linear optimization with hints where the cost function at round $t$ is $\|c_t\|_2 \le G$ and the hint $v_t$ is such that $c_t^\mathsf{T} v_t \ge \alpha \|c_t\|_2$, while $\|v_t\|_2 = 1$. Algorithm 1 in combination with $\mathcal{A}_{\mathrm{EXP}}$ has a worst-case regret of*

$$R(\mathcal{A}_{\mathrm{hint}}, c_{1:T}) \le \frac{d \cdot G \cdot R^2}{8\alpha \cdot C \cdot r} \cdot (1 + \log(T + 1)).$$

Since $\mathcal{A}_{\mathrm{EXP}}$ requires the coefficient of exp-concavity to be given as an input, $\alpha$ needs to be known a priori to be able to use Algorithm 1. However, we can use a standard doubling trick to relax this requirement and derive the same asymptotic regret bound. We defer the presentation of this argument to Appendix B.

# 4 Improved Regret Bounds for $(C, q)$-Uniformly Convex $\mathcal{K}$

In this section, we consider any feasible set $\mathcal{K}$ that is $(C, q)$-uniformly convex for $q \ge 2$. Our results differ from the previous section in two aspects. First, our algorithm can be used with $(C, q)$-uniformly convex feasible sets for any $q \ge 2$ compared to the results of the previous section that only hold for strongly convex sets ($q = 2$). On the other hand, the approach in this section requires the hints to be restricted to a finite set of vectors $\mathcal{V}$. We show that when $\mathcal{K}$ is $(C, q)$-uniformly convex for $q > 2$, our regret is $O(T^{\frac{2-q}{1-q}})$. If $q \in (2, 3)$, this is an improvement over the worst case regret of $O(\sqrt{T})$ guaranteed in the absence of hints.

We first consider the scenario where the hint is always pointing in the same direction, i.e. $v_t = v$ for some $v$ and all $t \in [T]$. In this case, we show how one can use a simple algorithm that picks the best performing action so far (a.k.a the Follow-The-Leader algorithm) to obtain improved regret bounds. We then consider the case where the hint belongs to a finite set $\mathcal{V}$. In this case, we instantiate one copy of the Follow-The-Leader algorithm for each $v \in \mathcal{V}$ and combine their outcomes in order to obtain improved regret bounds that depend on the cardinality of $\mathcal{V}$, which we denote by $|\mathcal{V}|$.

**Lemma 4.1.** *Suppose that $v_t = v$ for all $t = 1, \cdots, T$ and that $\mathcal{K}$ is $(C, q)$-uniformly convex that is symmetric around the origin, and $B_r \subseteq \mathcal{K} \subseteq B_R$ for some $r$ and $R$. Consider the algorithm, called Follow-The-Leader (FTL), that at every round $t$, plays $x_t \in \arg\min_{x \in \mathcal{K}} \sum_{\tau < t} c_\tau^\mathsf{T} x$. If $\sum_{\tau=1}^t c_\tau^\mathsf{T} v \ge 0$ for all $t = 1, \cdots, T$, then the regret is bounded as follows,*

$$R(\mathcal{A}_{\mathrm{FTL}}, c_{1:T}) \le \left( \frac{\|v\|_\mathcal{K} \cdot R^q}{2C} \right)^{1/(q-1)} \cdot \sum_{t=1}^T \left( \frac{\|c_t\|_2^q}{\sum_{\tau=1}^t c_\tau^\mathsf{T} v} \right)^{1/(q-1)}.$$

*Furthermore, when $v$ is a valid hint with margin $\alpha$, i.e., $c_t^\mathsf{T} v \ge \alpha \cdot \|c_t\|_2$ for all $t = 1, \cdots, T$, the right-hand side can be further simplified to obtain the regret bound:*

$$R(\mathcal{A}_{\mathrm{FTL}}, c_{1:T}) \le \frac{1}{2\gamma} \cdot G \cdot (\ln(T) + 1) \qquad \text{if } q = 2$$

*and*

$$R(\mathcal{A}_{\mathrm{FTL}}, c_{1:T}) \le \frac{1}{(2\gamma)^{1/(q-1)}} \cdot G \cdot \frac{q-1}{q-2} \cdot T^{\frac{q-2}{q-1}} \qquad \text{if } q > 2,$$

*where $\gamma = \frac{C \cdot \alpha}{\|v\|_\mathcal{K} \cdot R^q}$.*

*Proof.* We use a well-known inequality, known as FT(R)L Lemma (see e.g., [12, 17]), on the regret incurred by the FTL algorithm:

$$R(\mathcal{A}_{\mathrm{FTL}}, c_{1:T}) \le \sum_{t=1}^T c_t^\mathsf{T}(x_t - x_{t+1}).$$

Without loss of generality, we can assume that $\|x_t\|_\mathcal{K} = \|x_{t+1}\|_\mathcal{K} = 1$ since the maximum of a linear function is attained at a boundary point. Since $\mathcal{K}$ is $(C, q)$-uniformly convex, we have

$$\left\| \frac{x_t + x_{t+1}}{2} \right\|_\mathcal{K} \le 1 - \delta_\mathcal{K}(\|x_t - x_{t+1}\|_\mathcal{K}).$$

This implies that

$$\left\| \frac{x_t + x_{t+1}}{2} - \delta_{\mathcal{K}}(\|x_t - x_{t+1}\|_{\mathcal{K}}) \frac{v}{\|v\|_{\mathcal{K}}} \right\|_{\mathcal{K}} \le 1.$$

Moreover, $x_{t+1} \in \arg\min_{x \in \mathcal{K}} x^\top \sum_{\tau=1}^t c_\tau$. So, we have

$$\left( \sum_{\tau=1}^t c_\tau \right)^\top \left( \frac{x_t + x_{t+1}}{2} - \delta_{\mathcal{K}}(\|x_t - x_{t+1}\|_{\mathcal{K}}) \frac{v}{\|v\|_{\mathcal{K}}} \right) \ge \inf_{x \in \mathcal{K}} x^\top \sum_{\tau=1}^t c_\tau = x_{t+1}^\top \sum_{\tau=1}^t c_\tau.$$

Rearranging this last inequality and using the fact that $\sum_{\tau=1}^t v^\top c_\tau \ge 0$, we obtain:

$$\left( \sum_{\tau=1}^t c_\tau \right)^\top \left( \frac{x_t - x_{t+1}}{2} \right) \ge \delta_{\mathcal{K}}(\|x_t - x_{t+1}\|_{\mathcal{K}}) \cdot \frac{\sum_{\tau=1}^t v^\top c_\tau}{\|v\|_{\mathcal{K}}} \ge \frac{C \cdot \|x_t - x_{t+1}\|_2^q}{\|v\|_{\mathcal{K}} \cdot R^q} \cdot \left( \sum_{\tau=1}^t v^\top c_\tau \right).$$

By definition of FTL, we have $x_t \in \arg\min_{x \in \mathcal{K}} x^\top \sum_{\tau=1}^{t-1} c_\tau$, which implies:

$$\left( \sum_{\tau=1}^{t-1} c_\tau \right)^\top \frac{x_{t+1} - x_t}{2} \ge 0.$$

Summing up the last two inequalities and setting $\gamma = \frac{C \cdot \alpha}{\|v\|_{\mathcal{K}} \cdot R^q}$, we derive:

$$c_t^\top \left( \frac{x_t - x_{t+1}}{2} \right) \ge \frac{\gamma}{\alpha} \cdot \left( \sum_{\tau=1}^t v^\top c_\tau \right) \cdot \|x_t - x_{t+1}\|_2^q \ge \frac{\gamma}{\alpha} \cdot \left( \sum_{\tau=1}^t v^\top c_\tau \right) \cdot \frac{(c_t^\top (x_t - x_{t+1}))^q}{\|c_t\|_2^q}.$$

Rearranging this last inequality and using the fact that $\sum_{\tau=1}^t v^\top c_\tau \ge 0$, we obtain:

$$|c_t^\top (x_t - x_{t+1})| \le \frac{1}{(2\gamma/\alpha)^{1/(q-1)}} \cdot \left( \frac{\|c_t\|_2^q}{\sum_{\tau=1}^t v^\top c_\tau} \right)^{1/(q-1)}. \tag{4}$$

Summing (4) over all $t$ completes the proof of the first claim. The regret bounds for when $v^\top c_t \ge \alpha \cdot \|c_t\|_2$ for all $t = 1, \cdots, T$ follow from the first regret bound. We defer this part of the proof to Appendix D.2. $\qquad\square$

Note that the regret bounds become $O(T)$ when $q \to \infty$. This is expected because $L_q$ balls are $q$-uniformly convex for $q \ge 2$ and converge to $L_\infty$ balls as $q \to \infty$ and it is well-known that Follow-The-Leader yields $\Theta(T)$ regret in online linear optimization when $\mathcal{K}$ is a $L_\infty$ ball.

Using the above lemma, we introduce an algorithm for online linear optimization with hints that belong to a set $\mathcal{V}$. In this algorithm, we instantiate one copy of the FTL algorithm for each possible direction of the hint. On round $t$, we invoke the copy of the algorithm that corresponds to the direction of the hint $v_t$, using the history of the game for rounds with hints in that direction. We show that the overall regret of this algorithm is no larger than the sum of the regrets of the individual copies.

---

**Algorithm 2 $\mathcal{A}_{\text{set}}$: SET-OF-HINTS**

---

For all $v \in \mathcal{V}$, let $T_v = \emptyset$.
For $t = 1, \ldots, T$,

    1. Play $x_t \in \arg\min_{x \in \mathcal{K}} \sum_{\tau \in T_{v_t}} c_\tau^\top x$ and receive $c_t$ as feedback.

    2. Update $T_{v_t} \leftarrow T_{v_t} \cup \{t\}$.

---

**Theorem 4.2.** *Suppose that $\mathcal{K} \subseteq \mathbb{R}^d$ is a $(C, q)$-uniformly convex set that is symmetric around the origin, and $B_r \subseteq \mathcal{K} \subseteq B_R$ for some $r$ and $R$. Consider online linear optimization with hints where the cost function at round $t$ is $\|c_t\|_2 \le G$ and the hint $v_t$ comes from a finite set $\mathcal{V}$ and is such that $c_t^\top v_t \ge \alpha \|c_t\|_2$, while $\|v_t\|_2 = 1$. Algorithm 2 has a worst-case regret of*

$$R(\mathcal{A}_{\text{set}}, c_{1:T}) \le |\mathcal{V}| \cdot \frac{R^2}{2C \cdot \alpha \cdot r} \cdot G \cdot (\ln(T) + 1), \qquad \text{if } q = 2,$$

*and*

$$R(\mathcal{A}_{\text{set}}, c_{1:T}) \le |\mathcal{V}| \cdot \left( \frac{R^q}{2C \cdot \alpha \cdot r} \right)^{1/(q-1)} \cdot G \cdot \frac{q-1}{q-2} \cdot T^{\frac{q-2}{q-1}} \qquad \text{if } q > 2.$$

*Proof.* We decompose the regret as follows:

$$R(\mathcal{A}_{\text{set}}, c_{1:T}) = \sum_{t=1}^{T} c_t^{\intercal} x_t - \inf_{x \in \mathcal{K}} \sum_{t=1}^{T} c_t^{\intercal} x \leq \sum_{v \in \mathcal{V}} \left\{ \sum_{t \in T_v} c_t^{\intercal} x_t - \inf_{x \in \mathcal{K}} \sum_{t \in T_v} c_t^{\intercal} x \right\}$$

$$\leq |\mathcal{V}| \cdot \max_{v \in \mathcal{V}} R(\mathcal{A}_{\text{FTL}}, c_{T_v}).$$

The proof follows by applying Lemma 4.1 and by using $\|v_t\|_{\mathcal{K}} \leq (1/r) \cdot \|v_t\|_2 = 1/r$.

$\square$

Note that $\mathcal{A}_{\text{set}}$ does not require $\alpha$ or $\mathcal{V}$ to be known a priori, as it can compile the set of hint directions as it sees new ones. Moreover, if the hints are not limited to finite set $\mathcal{V}$ a priori, then the algorithm can first discretize the $L_2$ unit ball with an $\alpha/2$-net and approximate any given hint with one of the hints in the discretized set. Using this discretization technique, Theorem 4.2 can be extended to the setting where the hints are not constrained to a finite set while having a regret that is linear in the size of the $\alpha/2$-net (exponential in the dimension $d$.) Extensions of Theorem 4.2 are discussed in more details in the Appendix C.

# 5   Lower Bounds

The regret bounds derived in Sections 3 and 4 suggest that the curvature of $\mathcal{K}$ can make up for the lack of curvature of the loss function to get rates faster than $O(\sqrt{T})$ in online convex optimization, provided we receive additional information about the next move of the adversary in the form of a hint. In this section, we show that the curvature of the player's decision set $\mathcal{K}$ is necessary to get rates better than $O(\sqrt{T})$, even in the presence of a hint.

As an example, consider the unit cube, i.e. $\mathcal{K} = \{x \mid \|x\|_{\infty} \leq 1\}$. Note that this set is not uniformly convex. Since, the $i^{th}$ coordinate of points in such a set, namely $x_i$, has no effect on the range of acceptable values for the other coordinates, revealing one coordinate does not give us any information about the other coordinates $x_j$ for $j \neq i$. For example, suppose that $c_t$ has each of its first two coordinates set to $+1$ or $-1$ with equal probability and all other coordinates set to 1. In this case, even after observing the last $d - 2$ coordinates of the loss vector, the problem is reduced to a standard online linear optimization problem in the 2-dimensional unit cube. This choice of $c_t$ is known to incur a regret of $\Omega(\sqrt{T})$ [1]. Therefore, online linear optimization with the set $\mathcal{K} = \{x \mid \|x\|_{\infty} \leq 1\}$, even in the presence of hints, has a worst-case regret of $\Omega(\sqrt{T})$. As it turns out, this result holds for any polyhedral set of actions. We prove this by means of a reduction to the lower bounds established in [8] that apply to the online convex optimization framework (without hint). We defer the proof to the Appendix D.4.

**Theorem 5.1.** *If the set of feasible actions is a polyhedron then, depending on the set $\mathcal{C}$, either there exists a trivial algorithm that achieves zero regret or every online algorithm has worst-case regret $\Omega(\sqrt{T})$. This is true even if the adversary is restricted to pick a fixed hint $v_t = v$ for all $t = 1, \cdots, T$.*

At first sight, this result may come as a surprise. After all, since any $L_p$ ball with $1 < p \leq 2$ is strongly convex, one can hope to use a $L_{1+\nu}$ unit ball $\mathcal{K}'$ to approximate $\mathcal{K}$ when $\mathcal{K}$ is a $L_1$ ball (which is a polyhedron) and apply the results of Section 3 to achieve better regret bounds. The problem with this approach is that the constant in the modulus of convexity of $\mathcal{K}'$ deteriorates when $p \to 1$ since $\delta_{L_p}(\epsilon) = (p - 1) \cdot \epsilon^2$, see [3]. As a result, the regret bound established in Theorem 3.3 becomes $O(\frac{1}{p-1} \cdot \log T)$. Since the best approximation of a $L_1$ unit ball using a $L_p$ ball is of the form $\{x \in \mathbb{R}^d \mid d^{1-\frac{1}{p}} \|x\|_p \leq 1\}$, the distance between the offline benchmark in the definition of regret when using $\mathcal{K}'$ instead of $\mathcal{K}$ can be as large as $(1 - d^{\frac{1}{p}-1}) \cdot T$, which translates into an additive term of order $(1 - d^{\frac{1}{p}-1}) \cdot T$ in the regret bound when using $\mathcal{K}'$ as a proxy for $\mathcal{K}$. Due to the inverse dependence of the regret bound obtained in Theorem 3.3 on $p - 1$, the optimal choice of $p = 1 + \tilde{O}(\frac{1}{\sqrt{T}})$ leads to a regret of order $\tilde{O}(\sqrt{T})$.

Finally, we conclude with a result that suggests that $O(\log(T))$ is, in fact, the optimal achievable regret when $\mathcal{K}$ is strongly convex in online linear optimization with a hint. We defer the proof to the Appendix D.4.

**Theorem 5.2.** *If $\mathcal{K}$ is a $L_2$ ball then, depending on the set $\mathcal{C}$, either there exists a trivial algorithm that achieves zero regret or every online algorithm has worst-case regret $\Omega(\log(T))$. This is true even if the adversary is restricted to pick a fixed hint $v_t = v$ for all $t = 1, \cdots, T$.*

## 6  Directions for Future Research

We conjecture that the dependence of our regret bounds with respect to $T$ is suboptimal when $\mathcal{K}$ is $(C, q)$-uniformly convex for $q > 2$. We expect the optimal rate to converge to $\sqrt{T}$ when $q \to \infty$ as $L_q$ balls converge to $L_\infty$ balls and it is well known that the minimax regret scales as $\sqrt{T}$ in online linear optimization without hints when the decision set is a $L_\infty$ ball. However, this calls for the development of an algorithm that is not based on a reduction to the Follow-The-Leader algorithm, as discussed after Lemma 4.1.

We also conjecture that it is possible to relax the assumption that there are finitely many hints when $\mathcal{K}$ is $(C, q)$-uniformly convex with $q > 2$ without incurring an exponential dependence of the regret bounds (and the runtime) on the dimension $d$, see Appendix C. Again, this calls for the development of an algorithm that is not based on a reduction to the Follow-The-Leader algorithm.

A solution that would alleviate the two aforementioned shortcomings would likely be derived through a reduction to online convex optimization with convex functions that are $(C, q)$-uniformly convex, for $q \geq 2$, in all but one direction and constant in the other, in a similar fashion as done in Section 3 when $q = 2$. There has been progress in this direction in the literature, but, to the best of our knowledge, no conclusive result yet. For instance, Vovk [23] studies a related problem but restricts the study to the squared loss function. It is not clear if the setting studied in this paper can be reduced to the setting of square loss function. Another example is given by [21], where the authors consider online convex optimization with general $(C, q)$-uniformly convex functions in Banach spaces (with no hint) achieving a regret of order $O(T^{(q-2)/(q-1)})$. Note that this rate matches the one derived in Theorem 4.2. However, as noted above, our setting cannot be reduced to theirs because our virtual loss functions are not uniformly convex in every direction.

### Acknowledgments

Haghtalab was partially funded by an IBM Ph.D. fellowship and a Microsoft Ph.D. fellowship. Jaillet acknowledges the research support of the Office of Naval Research (ONR) grant N00014-15-1-2083. This work was partially done when Haghtalab was an intern at Microsoft Research, Redmond WA.

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
