[Supplementary Material · hints_NIPS17_supp.pdf]

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

# A Additional Related Works

The notion of hint introduced in this work is quite general and arises naturally in a variety of settings. Indeed, this notion generalizes some of the previous notions of predictability in online convex optimization.

Aside from the example, mentioned in Section 1.1, where a hint on the first coordinate of the loss vector is provided to the player, Hazan and Megiddo [13] also considered modeling the prior information in each round as a state space and measuring regret against a stronger benchmark that uses a mapping from the state space to the feasible set. Chiang and Lu [6] considered actively querying bits of the loss vector, but their result mainly improves the dependence of the regret on the dimension $d$.

Another notion of predictability is concerned with predictability of the entire loss vector rather than individual bits. Chiang et al. [7] considered online convex optimization with a sequence of loss functions that demonstrates a gradual change and they derived a regret bound in terms of the deviation of the loss functions. Rakhlin and Sridharan [19, 20] extended this line of work beyond sequences with gradual changes and showed that one can achieve an improved regret bound if the gradient of the loss function is predictable. They also applied this method to offline optimization problems such as Max Flow and to the problem of computing Nash equilibria in zero-sum games. In the latter case, they showed that when both players employ a variant of the Mirror Prox algorithm, they converge to the minimax equilibrium at a rate $O(\log(T))$. We compare our results to results derived in the literature on online convex optimization in more details below.

**Comparison with [13]**  Hazan and Megiddo [13] considered as an example a setting where the player knows the first coordinate of the loss vector at all rounds, and showed that when $|c_{t1}| > 0$ and when the set of feasible actions is the Euclidean ball, one can achieve a regret of $O(d \cdot \log(T))$. Our work directly improves over this result, as in our setting a hint $v_t = \pm e_1$ also achieves $O(\log(T))$ regret, where the hidden factors are independent of $d$ (see Theorem 4.2 with $\mathcal{V} = \{e_1, -e_1\}$). Moreover, we can deal with hints in different directions at different rounds and we allow for general uniformly convex action sets.

**Connection with [19, 20]**  Suppose that we are provided with a vector $\tilde{c}_t$ at the beginning of time period $t$ such that $\tilde{c}_t$ approximates $c_t$ in the following sense: $\|c_t - \tilde{c}_t\|_2 \leq \sigma\|c_t\|_2$ for $\sigma \in [0, 1)$ (if $\sigma \geq 1$ this gives us essentially no information because $\tilde{c}_t = 0$ is a valid choice). Then $\frac{\tilde{c}_t}{\|\tilde{c}_t\|_2}$ (or 0 if $\tilde{c}_t = 0$) is a valid hint with margin $\alpha = (1-\sigma)/(1+\sigma)$. Indeed note that $\|\tilde{c}_t\|_2 \leq (1+\sigma) \cdot \|c_t\|_2$ and $\|\tilde{c}_t\|_2 \geq (1-\sigma) \cdot \|c_t\|_2$. Moreover:

$$\frac{\tilde{c}_t^\top c_t}{\|\tilde{c}_t\|_2} = \frac{1}{2} \cdot \frac{\|c_t\|_2^2 + \|\tilde{c}_t\|_2^2 - \|\tilde{c}_t - c_t\|_2^2}{\|\tilde{c}_t\|_2}$$

$$\geq \frac{1}{2} \cdot (1 + (1-\sigma)^2 - \sigma^2) \cdot \frac{\|c_t\|_2^2}{\|\tilde{c}_t\|_2}$$

$$= (1-\sigma) \cdot \frac{\|c_t\|_2}{\|\tilde{c}_t\|_2}$$

$$\geq \frac{1-\sigma}{1+\sigma} \cdot \|c_t\|_2.$$

When $\mathcal{K}$ is strongly convex (e.g. a $L_2$ ball) and $\|c_t - \tilde{c}_t\|_2 \leq \sigma\|c_t\|_2$ for $\sigma \in [0, 1)$ for all time periods $t = 1, \cdots, T$, we get a regret bound of order $O((1+\sigma)/(1-\sigma) \cdot \log(T))$ which improves upon the bound $\sigma \cdot \sqrt{\sum_{t=1}^T \|c_t\|_2^2}$ obtained in [19]. However, the regret bounds that we get are not adaptive, i.e. we need to assume that $\|c_t - \tilde{c}_t\|_2 \leq \sigma\|c_t\|_2$ holds at all time periods to establish the regret bounds. This is in contrast with the analysis carried out in [19] where the regret bounds adapt to the sequence $(c_1, \cdots, c_T)$ at hand.

**Comparison with [15]**  The regret bounds obtained for the Follow-The-Leader algorithm in Lemma 4.1 are incomparable to the ones obtained by Huang et al. [15] for online linear optimization (without hints) in the sense that: (i) their bounds adapt to the sequence of cost vectors $(c_1, \cdots, c_T)$ without

making any assumption about them a priori whereas, in our case, the hints are assumed to be valid a priori, and (ii) our results hold for general uniformly convex sets, which enables us to get intermediate rates that interpolate between $\log(T)$ and $\sqrt{T}$, whereas their results hold for stronger convex sets.

## B    Doubling Trick when $\mathcal{K}$ is Strongly Convex and $\alpha$ is not Known a Priori.

We break down the horizon into phases where, during phase $i \in \mathbb{N}$, we run $\mathcal{A}_{\text{hint}}$ from scratch (discarding all previously observed values of the loss vectors $c_t$ and the hints $v_t$) with exp-concavity parameter taken as $2^{i-1} \cdot \frac{8 \cdot C \cdot r}{G \cdot R^2}$. Phase $i+1$ begins at time $t_{i+1}$ when the hint is no longer valid with margin $1/2^{i-1}$, i.e.:

$$t_{i+1} = \min\{\tau \geq t_i \mid c_\tau^\mathsf{T} v_\tau < (1/2^{i-1}) \cdot \|c_\tau\|_2\} + 1$$

(with the convention that $0/0 = 1$) and phase 1 begins at time $t_1 = 1$.

**Lemma B.1.** *When $\alpha$ is not known a priori, using the doubling trick yields a regret bound that is identical (up to a constant additive term) to the one we would obtain if we knew $\alpha$ a priori. Specifically, we have:*

$$R(\mathcal{A}_{\text{hint}}, c_{1:T}) \leq 2 \log(\frac{1}{\alpha}) \cdot G \cdot R + \frac{d \cdot G \cdot R^2}{8\alpha \cdot C \cdot r} \cdot (1 + \log(1 + T)).$$

*Proof.* Let $N$ denote the number of phases and define $t_{N+1} = T + 1$. Note that there are at most $N \leq \log(1/\alpha)$ phases since $\frac{c_t^\mathsf{T} v_t}{\|c_t\|_2} \in [\alpha, 1]$ for all time periods $t$ by assumption. Observe that, for any phase $i$ and for any time period $t = t_i, \cdots, t_{i+1} - 2$, we have $\frac{c_t^\mathsf{T} v_t}{\|c_t\|_2} \geq 1/2^{i-1}$ so that $\hat{c}_t(\cdot)$ is $2^{i-1} \cdot \frac{8C \cdot r}{G \cdot R^2}$-exp-concave. Using Lemmas 3.1 and 3.2, we get:

$$\sum_{t=1}^{T} c_t^\mathsf{T} x_t - \inf_{x \in \mathcal{K}} \sum_{t=1}^{T} c_t^\mathsf{T} x \leq \sum_{i=1}^{N} \{ \sum_{t=t_i}^{t_{i+1}-1} c_t^\mathsf{T} x_t - \inf_{x \in \mathcal{K}} \sum_{t=t_i}^{t_{i+1}-1} c_t^\mathsf{T} x \}$$

$$\leq 2N \cdot G \cdot R + \sum_{i=1}^{N} \{ \sum_{t=t_i}^{t_{i+1}-2} c_t^\mathsf{T} x_t - \inf_{x \in \mathcal{K}} \sum_{t=t_i}^{t_{i+1}-2} c_t^\mathsf{T} x \}$$

$$\leq 2 \log(\frac{1}{\alpha}) \cdot G \cdot R + \sum_{i=1}^{N} 2^{i-1} \cdot \frac{d \cdot G \cdot R^2}{8C \cdot r} \cdot (1 + \log(1 + t_{i+1} - t_i - 1))$$

$$\leq 2 \log(\frac{1}{\alpha}) \cdot G \cdot R + 2^{N} \cdot \frac{d \cdot G \cdot R^2}{8C \cdot r} \cdot (1 + \log(1 + T))$$

$$\leq 2 \log(\frac{1}{\alpha}) \cdot G \cdot R + \frac{d \cdot G \cdot R^2}{8\alpha \cdot C \cdot r} \cdot (1 + \log(1 + T)),$$

where we use $\|c_t\|_2 \leq G$ and $\|x_t\|_2 \leq R \cdot \|x_t\|_{\mathcal{K}} = R$ for the second inequality. This concludes the proof. $\square$

## C    Extensions of Theorem 4.2.

**Hints pointing in arbitrary directions**    When the directions of the hints are arbitrary, we can discretize the $L_2$ unit sphere using an $\alpha/2$-net (which contains at most $(1 + \frac{4}{\alpha})^n$ points, see [22]), which we denote by $\tilde{\mathcal{V}}$. At any time $t = 1, \cdots, T$, the hint $v_t$ is first mapped to its closest neighbor in $\tilde{\mathcal{V}}$, denoted by $\tilde{v}_t$, and we use $\mathcal{A}_{\text{set}}$ with $\mathcal{V} = \tilde{\mathcal{V}}$ and $\tilde{v}_t$ as "the" hint. We refer to this new algorithm as $\tilde{\mathcal{A}}_{\text{set}}$.

**Theorem C.1.** *Suppose that $\mathcal{K}$ is $(C, q)$-uniformly convex. If the hints come from a finite set $\mathcal{V}$, then $\tilde{\mathcal{A}}_{\text{set}}$ yields the same regret bounds as $\mathcal{A}_{\text{set}}$ up to a multiplicative factor 2. If the hints point in arbitrary directions then we have:*

$$R(\tilde{\mathcal{A}}_{\text{set}}, c_{1:T}) \leq (1 + \frac{4}{\alpha})^d \cdot \frac{R^2}{C \cdot \alpha \cdot r} \cdot G \cdot (\ln(T) + 1), \qquad \text{if } q = 2,$$

*and*

$$R(\tilde{\mathcal{A}}_{\text{set}}, c_{1:T}) \leq (1 + \tfrac{4}{\alpha})^d \cdot \left( \frac{R^q}{C \cdot \alpha \cdot r} \right)^{1/(q-1)} \cdot G \cdot \frac{q-1}{q-2} \cdot T^{\frac{q-2}{q-1}} \qquad \text{if } q > 2.$$

*Proof.* Observe that, at any time period $t = 1, \cdots, T$, $\tilde{v}_t$ is a valid hint with margin $\alpha/2$. Indeed:

$$\begin{aligned}
c_t^{\mathsf{T}} \tilde{v}_t &= c_t^{\mathsf{T}} v_t - c_t^{\mathsf{T}}(v_t - \tilde{v}_t) \\
&\geq \alpha \cdot \|c_t\|_2 - \|c_t\|_2 \cdot \|v_t - \tilde{v}_t\|_2 \\
&\geq \alpha/2 \cdot \|c_t\|_2,
\end{aligned}$$

by definition of $\tilde{\mathcal{V}}$. If the hints come from a finite set $\mathcal{V}$, the set of hints $\{\tilde{v}_t \mid t = 1, \cdots, T\}$ is also finite with cardinality at most $|\mathcal{V}|$ since the mapping $v_t \to \tilde{v}_t$ is independent of $t$. This shows the first part of the claim. If the hints point in arbitrary directions, then let, for each $v \in \tilde{\mathcal{V}}$, $T_v$ be the subset of time periods such that $v_t$ is mapped to $v$. We have:

$$\begin{aligned}
R(\tilde{\mathcal{A}}_{\text{set}}, c_{1:T}) &= \sum_{t=1}^{T} c_t^{\mathsf{T}} x_t - \inf_{x \in \mathcal{K}} \sum_{t=1}^{T} c_t^{\mathsf{T}} x \\
&\leq \sum_{v \in \tilde{\mathcal{V}}} \{ \sum_{t \in T_v} c_t^{\mathsf{T}} x_t - \inf_{x \in \mathcal{K}} \sum_{t \in T_v} c_t^{\mathsf{T}} x \} \\
&\leq |\tilde{\mathcal{V}}| \cdot \max_{v \in \tilde{\mathcal{V}}} R(\mathcal{A}_{\text{FTL}}, c_{T_v}),
\end{aligned}$$

which concludes the proof with Lemma 4.1. $\qquad\qquad\square$

**Random hints**   We consider an extension to a stochastic setting where the hint is not necessarily always valid at each round but rather in expectations. We next show that $\mathcal{A}_{\text{set}}$ yields regret bounds similar to the ones derived when the hint is always valid.

**Theorem C.2.** *Suppose that: (a) the hints come from a finite set $\mathcal{V}$, (b) $\mathcal{K}$ is $(C, q)$-uniformly convex, (c) $((c_t, v_t))_{t \in \mathbb{N}}$ is an independent stochastic process (but not necessarily i.i.d.), and (d) $\mathbb{E}[c_t^{\mathsf{T}} v_t \mid v_t = v] \geq \alpha \cdot \mathbb{E}[\|c_t\|_2 \mid v_t = v]$ for all $v \in \mathcal{V}$. Then, we have:*

$$\mathbb{E}[R(\mathcal{A}_{\text{set}}, c_{1:T})] \leq |\mathcal{V}| \cdot \frac{7R^2}{C \cdot \alpha \cdot r} \cdot G \cdot (\ln(T) + 1) + O(1),$$

*if $q = 2$ and*

$$\mathbb{E}[R(\mathcal{A}_{\text{set}}, c_{1:T})] \leq |\mathcal{V}| \cdot \left( \frac{7R^q}{C \cdot \alpha \cdot r} \right)^{1/(q-1)} \cdot G \cdot \frac{q-1}{q-2} \cdot T^{\frac{q-2}{q-1}} + O(1),$$

*if $q > 2$.*

The proof is deferred to the Appendix D.3. Note that Theorem C.2 is a strict generalization of Theorem 4.2 since we allow the sequence $((c_t, v_t))_{t \in \mathbb{N}}$ not to be identically distributed.

To illustrate the applicability of Theorem C.2, we consider the setting studied in [15] where $((c_t))_{t \in \mathbb{N}}$ is an i.i.d. stochastic process with mean $\mu \neq 0$ and no hint is available. In this setting, we can take $v_t = \mu / \|\mu\|_2$ as the hint at any time period for the purpose of the analysis (but $\mu$ need not be known since we are using a single instance of FTL when the hint is the same at all time periods). Indeed, in this case, we have:

$$\begin{aligned}
\mathbb{E}[c_t^{\mathsf{T}} v_t] &= \mathbb{E}[c_t]^{\mathsf{T}} \frac{\mu}{\|\mu\|_2} \\
&= \|\mu\|_2 \\
&\geq \frac{\|\mu\|_2}{G} \cdot \mathbb{E}[\|c_t\|_2]
\end{aligned}$$

as long as $\|c_t\|_2 \leq G$. Hence, when the set $\mathcal{K}$ is a $L_2$ ball, we recover the regret bound $O(\frac{G^2}{\|\mu\|_2} \cdot \log(T))$ established in [15].

# D  Omitted Proofs

## D.1  Omitted Proof for Lemma 3.1

**Lemma D.1.** *The function $\hat{c}_t(\cdot)$ is continuous for any $t \in \mathbb{N}$.*

*Proof.* Take a point $x \in \mathcal{K}$ and a sequence $x_n \in \mathcal{K} \to x$. For every $n$, there exists $w_n \in \mathcal{K}$ such that $w_n^\perp = x_n^\perp$ and $\hat{c}_t(x_n) = c_t^\mathsf{T} w_n$. Observe that the sequence $(c_t^\mathsf{T} w_n)_{n \in \mathbb{N}}$ is bounded since $\|w_n\|_{\mathcal{K}} \le 1$. Hence, it is sufficient to show that $(c_t^\mathsf{T} w_n)_{n \in \mathbb{N}}$ has a unique limit point $\hat{c}_t(x)$. Consider a subsequence of $(c_t^\mathsf{T} w_n)_{n \in \mathbb{N}}$ that converges and, since $\mathcal{K}$ is compact, extract further a subsequence of $(w_n)_{n \in \mathbb{N}}$ that converges to some $w_\infty \in \mathcal{K}$. Without loss of generality, we continue to assume that these sequences are indexed by all $n \in \mathbb{N}$. Taking limits in $w_n^\perp = x_n^\perp$, we get $w_\infty^\perp = x^\perp$ (an orthogonal projection is a linear operator and thus is continuous in finite dimension). Consider $w \in \mathcal{K}$ such that $\|w\|_{\mathcal{K}} < 1$ and $w^\perp = x^\perp$. For $n$ big enough we have $\left\|x^\perp - x_n^\perp\right\|_{\mathcal{K}} \le 1 - \|w\|_{\mathcal{K}}$ and so $\left\|w + x_n^\perp - x^\perp\right\|_{\mathcal{K}} \le 1$ and $(w + x_n^\perp - x^\perp)^\perp = x_n^\perp$. By definition of $w_n$, we get $c_t^\mathsf{T} w_n^\perp \le c_t^\mathsf{T} w + c_t^\mathsf{T}(x_n^\perp - x^\perp)$. Taking limits, we get $c_t^\mathsf{T} w_\infty \le c_t^\mathsf{T} w$ for all $w$ such that $\|w\|_{\mathcal{K}} < 1$ and $w^\perp = x^\perp$. By continuity of linear functions this also holds for all $w \in \mathcal{K}$ such that $w^\perp = x^\perp$ and we get $\hat{c}_t(x) = c_t^\mathsf{T} w_\infty = \lim_{n \to \infty} c_t^\mathsf{T} w_n = \lim_{n \to \infty} \hat{c}_t(x_n)$. $\square$

## D.2  Omitted Proofs for Lemma 4.1

As we showed in the main body of the paper, we have:

$$|c_t^\mathsf{T}(x_t - x_{t+1})| \le \frac{1}{(2\gamma/\alpha)^{1/(q-1)}} \cdot \left( \frac{\|c_t\|_2^q}{\sum_{\tau=1}^t v^\mathsf{T} c_\tau} \right)^{1/(q-1)}.$$

In what follows, we further assume that $v$ is a valid hint with margin $\alpha$, i.e., $v^\mathsf{T} c_t \ge \alpha \cdot \|c_t\|_2$ for all $t = 1, \cdots, T$. We get:

$$|c_t^\mathsf{T}(x_t - x_{t+1})| \le \frac{1}{(2\gamma)^{1/(q-1)}} \cdot \left( \frac{\|c_t\|_2^q}{\sum_{\tau=1}^t \|c_\tau\|_2} \right)^{1/(q-1)}.$$

Note that the right-hand side is finite even if $c_t = 0$. Plugging this last inequality back into the first regret bound, we derive:

$$R(\mathcal{A}_{\mathrm{FTL}}, c_{1:T}) \le \frac{1}{(2\gamma)^{1/(q-1)}} \cdot \sum_{t=1}^T \left( \frac{\|c_t\|_2^q}{\sum_{\tau=1}^t \|c_\tau\|_2} \right)^{1/(q-1)}$$

$$\le \frac{1}{(2\gamma)^{1/(q-1)}} \cdot \sup_{(y_1, \cdots, y_T) \in [0,G]^T} \sum_{t=1}^T \left( \frac{y_t^q}{\sum_{\tau=1}^t y_\tau} \right)^{1/(q-1)}.$$

We prove below that for any $t = 1, \cdots, T$ and any fixed values $(y_1, \cdots, y_{t-1}, y_{t+1}, \cdots, y_T) \in [0,G]^{T-1}$, the function $y_t \to \sum_{n=1}^T \left( \frac{y_n^q}{\sum_{\tau=1}^n y_\tau} \right)^{1/(q-1)}$ is convex on $[0, G]$ and thus the maximum of this function is attained at an extreme point: either $0$ or $G$. Repeating this process for $t = 1, \cdots, T$, we get:

$$R(\mathcal{A}_{\mathrm{FTL}}, c_{1:T}) \le \frac{1}{(2\gamma)^{1/(q-1)}} \cdot \sup_{(y_1, \cdots, y_T) \in \{0,G\}^T} \sum_{t=1}^T \left( \frac{y_t^q}{\sum_{\tau=1}^t y_\tau} \right)^{1/(q-1)}$$

$$\le \frac{1}{(2\gamma)^{1/(q-1)}} \cdot \sup_{n=0, \cdots, T} \sum_{k=1}^n \left( \frac{G^q}{k \cdot G} \right)^{1/(q-1)}$$

$$\le \frac{1}{(2\gamma)^{1/(q-1)}} \cdot G \cdot \sum_{k=1}^T \frac{1}{k^{1/(q-1)}}.$$

This concludes the proof as $\sum_{k=1}^{T}\frac{1}{k^{1/(q-1)}} \leq \ln(T)+1$ if $q=2$ and $\sum_{k=1}^{T}\frac{1}{k^{1/(q-1)}} \leq \frac{q-1}{q-2}\cdot T^{\frac{q-2}{q-1}}$ if $q>2$.

Let us now prove that, for any $t=1,\cdots,T$ and any fixed values $(y_1,\cdots,y_{t-1},y_{t+1},\cdots,y_T) \in [0,G]^{T-1}$, the function $y_t \rightarrow \sum_{n=1}^{T}(\frac{y_n^q}{\sum_{\tau=1}^{n}y_\tau})^{1/(q-1)}$ is convex on $[0,G]$. Clearly, $y_t \rightarrow \sum_{n\neq t}(\frac{y_n^q}{\sum_{\tau=1}^{n}y_\tau})^{1/(q-1)}$ is convex on $[0,G]$ since $y_t$ only appears in the denominator and $1/(q-1) \geq 0$. We use the shorthand $A=\sum_{\tau=1}^{t-1}y_\tau$. It remains to show that $\phi : y \rightarrow (\frac{y^q}{y+A})^{1/(q-1)}$ is convex. We have:

$$\phi''(y)=\frac{q}{(q-1)^2}\cdot y^{q/(q-1)-2}\cdot A^2\cdot(y+A)^{-1/(q-1)-2},$$

which is non-negative for $A \geq 0$ and $y > 0$ (for $y=0$, we can directly show by hand that $\phi(\lambda\cdot 0+(1-\lambda)\cdot z) \leq \lambda\cdot\phi(0)+(1-\lambda)\cdot\phi(z)$ for $\lambda\in[0,1]$ and $z\geq 0$).

### D.3 Proof of Theorem C.2

We will need the following concentration inequality.

**Lemma D.2** (Chernoff-Hoeffding concentration inequality). *Let $(X_t)_{t=1,\cdots,T}$ be a sequence of jointly independent random variables in $[0,1]$. We have, for any $\epsilon\in(0,1)$:*

$$\mathbb{P}\left[\sum_{t=1}^{T}X_t > (1+\epsilon)\cdot\sum_{t=1}^{T}\mathbb{E}[X_t]\right] \leq \exp(-\frac{\epsilon^2}{3}\cdot\sum_{t=1}^{T}\mathbb{E}[X_t])$$

*and*

$$\mathbb{P}\left[\sum_{t=1}^{T}X_t < (1-\epsilon)\cdot\sum_{t=1}^{T}\mathbb{E}[X_t]\right] \leq \exp(-\frac{\epsilon^2}{2}\cdot\sum_{t=1}^{T}\mathbb{E}[X_t]).$$

To establish the claims of Theorem C.2, first observe that:

$$\mathbb{E}[R(\mathcal{A}_{\text{set}},c_{1:T})] = \mathbb{E}[\sum_{t=1}^{T}c_t^\intercal x_t - \inf_{x\in\mathcal{K}}\sum_{t=1}^{T}c_t^\intercal x]$$
$$\leq \sum_{v\in\mathcal{V}}\{\mathbb{E}[\sum_{t\in T_v}c_t^\intercal x_t - \inf_{x\in\mathcal{K}}\sum_{t\in T_v}c_t^\intercal x]\}$$
$$\leq |\mathcal{V}|\cdot\max_{v\in\mathcal{V}}\mathbb{E}[R(\mathcal{A}_{\text{FTL}},c_{T_v})].$$

Take $v\in\mathcal{V}$. In the same spirit as in Lemma 4.1, what remains to be done is to bound the regret incurred by Follow-The-Leader for any independent stochastic sequence $(c_t)_{t\in\mathbb{N}}$ such that $\mathbb{E}[c_t^\intercal v] \geq \alpha\cdot\mathbb{E}[\|c_t\|_2]$ for all $t=1,\cdots,T$. We start with the same standard inequality on the regret incurred by Follow-The-Leader:

$$\mathbb{E}[R(\mathcal{A}_{\text{FTL}},c_{1:T})] \leq \mathbb{E}[\sum_{t=1}^{T}c_t^\intercal(x_t-x_{t+1})]. \tag{5}$$

Without loss of generality, we can assume that $\|x_t\|_\mathcal{K}=\|x_{t+1}\|_\mathcal{K}=1$. Hence, we have:

$$\left\|\frac{x_t+x_{t+1}}{2}\right\|_\mathcal{K} \leq 1-\delta_\mathcal{K}(\|x_t-x_{t+1}\|_\mathcal{K}),$$

which implies that

$$\left\|\frac{x_t+x_{t+1}}{2}-\frac{6}{7}\cdot\delta_\mathcal{K}(\|x_t-x_{t+1}\|_\mathcal{K})\cdot rv\right\|_\mathcal{K} \leq 1,$$

since $\|v\|_\mathcal{K} \leq 1/r\cdot\|v\|_2=1/r$. Since moreover we have $x_{t+1}\in\arg\min_{x\in\mathcal{K}}x^\intercal\sum_{\tau=1}^{t}c_\tau$, we derive, along the same lines as in Lemma 4.1, that:

$$\left(\sum_{\tau=1}^{t}c_\tau\right)^\intercal\frac{x_t-x_{t+1}}{2} \geq \frac{6}{7}\cdot\delta_\mathcal{K}(\|x_t-x_{t+1}\|_\mathcal{K})\cdot r\cdot\sum_{\tau=1}^{t}v^\intercal c_\tau \tag{6}$$

Moreover, since $x_t \in \arg\min_{x \in \mathcal{K}} x^{\mathsf{T}} \sum_{\tau=1}^{t-1} c_\tau$, we have:

$$\left(\sum_{\tau=1}^{t-1} c_\tau\right)^{\mathsf{T}} \frac{x_{t+1} - x_t}{2} \geq 0.$$

Summing these two inequalities yields:

$$c_t^{\mathsf{T}} \frac{x_t - x_{t+1}}{2} \geq \frac{6}{7} \cdot \delta_{\mathcal{K}}(\|x_t - x_{t+1}\|_{\mathcal{K}}) \cdot r \cdot \sum_{\tau=1}^{t} v^{\mathsf{T}} c_\tau.$$

Similarly, since:

$$\left\| \frac{x_t + x_{t+1}}{2} \pm \delta_{\mathcal{K}}(\|x_t - x_{t+1}\|_{\mathcal{K}}) \frac{c_t}{\|c_t\|_{\mathcal{K}}} \right\|_{\mathcal{K}} \leq 1,$$

$x_{t+1} \in \arg\min_{x \in \mathcal{K}} x^{\mathsf{T}} \sum_{\tau=1}^{t} c_\tau$, and $x_t \in \arg\min_{x \in \mathcal{K}} x^{\mathsf{T}} \sum_{\tau=1}^{t-1} c_\tau$, we get:

$$(\sum_{\tau=1}^{t} c_\tau)^{\mathsf{T}} \frac{x_t - x_{t+1}}{2} \geq \delta_{\mathcal{K}}(\|x_t - x_{t+1}\|_{\mathcal{K}}) \cdot \frac{\sum_{\tau=1}^{t} c_t^{\mathsf{T}} c_\tau}{\|c_t\|_{\mathcal{K}}}$$

and

$$\left(\sum_{\tau=1}^{t-1} c_\tau\right)^{\mathsf{T}} \frac{x_{t+1} - x_t}{2} \geq \delta_{\mathcal{K}}(\|x_t - x_{t+1}\|_{\mathcal{K}}) \cdot \frac{\sum_{\tau=1}^{t-1} -c_t^{\mathsf{T}} c_\tau}{\|c_t\|_{\mathcal{K}}}.$$

Summing these last two inequalities yields:

$$\begin{aligned} c_t^{\mathsf{T}} \frac{x_t - x_{t+1}}{2} &\geq \delta_{\mathcal{K}}(\|x_t - x_{t+1}\|_{\mathcal{K}}) \cdot \frac{\|c_t\|_2^2}{\|c_t\|_{\mathcal{K}}} \\ &\geq \delta_{\mathcal{K}}(\|x_t - x_{t+1}\|_{\mathcal{K}}) \cdot r \cdot \|c_t\|_2, \end{aligned} \tag{7}$$

since $\|c_t\|_{\mathcal{K}} \leq 1/r \cdot \|c_t\|_2$. Define the events:

$$A_t = \{\sum_{\tau=1}^{t-1} v^{\mathsf{T}} c_\tau < \frac{1}{2} \cdot \sum_{\tau=1}^{t-1} \mathbb{E}[v^{\mathsf{T}} c_\tau]\}$$

and

$$B_t = \{\sum_{\tau=1}^{t-1} \|c_\tau\|_2 > \frac{3}{2} \cdot \sum_{\tau=1}^{t-1} \mathbb{E}[\|c_\tau\|_2]\}.$$

Using Lemma D.2, we have:

$$\begin{aligned} \mathbb{P}[A_t] &\leq \exp(-\frac{1}{8G} \cdot \sum_{\tau=1}^{t-1} \mathbb{E}[v^{\mathsf{T}} c_\tau]) \\ &\leq \exp(-\frac{\alpha}{16G} \cdot \sum_{\tau=1}^{t-1} \mathbb{E}[\|c_\tau\|_2]), \end{aligned}$$

and

$$\mathbb{P}[B_t] \leq \exp(-\frac{1}{12G} \cdot \sum_{\tau=1}^{t-1} \mathbb{E}[\|c_\tau\|_2]),$$

since $|v^{\mathsf{T}} c_t| \leq \|c_t\|_2 \leq G$. Note that, conditioned on the events $A_t^{\complement}$ and $B_t^{\complement}$, summing inequalities (6) and (7) yields:

$$\begin{aligned} c_t^{\mathsf{T}}(x_{t+1} - x_t) &\geq \delta_{\mathcal{K}}(\|x_t - x_{t+1}\|_{\mathcal{K}}) \cdot r \cdot \left(\|c_t\|_2 + 6/7 v^{\mathsf{T}} c_t + 6/7 \sum_{\tau=1}^{t-1} v^{\mathsf{T}} c_\tau\right) \\ &\geq \delta_{\mathcal{K}}(\|x_t - x_{t+1}\|_{\mathcal{K}}) \cdot r \cdot \left(1/7 \|c_t\|_2 + \alpha/7 \sum_{\tau=1}^{t-1} \|c_\tau\|_2\right) \end{aligned}$$

$$\geq \frac{C \cdot r \cdot \alpha}{7 \cdot R^q} \cdot \left( \sum_{\tau=1}^{t} \|c_\tau\|_2 \right) \cdot \|x_t - x_{t+1}\|_2^q.$$

Following the same steps as in Lemma 4.1, we conclude that:

$$|c_t^\mathsf{T}(x_t - x_{t+1})| \leq \frac{1}{(\gamma)^{1/(q-1)}} \cdot \left( \frac{\|c_t\|_2^q}{\sum_{\tau=1}^{t} \|c_\tau\|_2} \right)^{1/(q-1)},$$

with $\gamma = \frac{C \cdot r \cdot \alpha}{7R^q}$. Plugging this last inequality back into (5), we get:

$$\mathbb{E}[R(\mathcal{A}^{\mathrm{FTL}}, c_{1:T})] \leq \sum_{t=1}^{T} \mathbb{E}\left[\|c_t\|_2 \cdot (\|x_t\|_2 + \|x_{t+1}\|_2) \cdot 1_{A_t \cup B_t}\right]$$

$$+ \frac{1}{(\gamma)^{1/(q-1)}} \cdot \sum_{t=1}^{T} \mathbb{E}\left[ \left( \frac{\|c_t\|_2^q}{\sum_{\tau=1}^{t} \|c_\tau\|_2} \right)^{1/(q-1)} \cdot 1_{A_t^\mathsf{c} \cap B_t^\mathsf{c}} \right]$$

$$\leq 2R \cdot \sum_{t=1}^{T} \mathbb{E}[\|c_t\|_2 \cdot 1_{A_t \cup B_t}]$$

$$+ \frac{1}{(\gamma)^{1/(q-1)}} \cdot \sup_{p_1, \cdots, p_T \in \mathcal{P}(\mathcal{C})} \mathbb{E}\left[ \sum_{t=1}^{T} \left( \frac{\|c_t\|_2^q}{\sum_{\tau=1}^{t} \|c_\tau\|_2} \right)^{1/(q-1)} \right]$$

$$= 2R \cdot \sum_{t=1}^{T} \mathbb{E}[\|c_t\|_2] \cdot (\mathbb{P}[A_t] + \mathbb{P}[B_t])$$

$$+ \frac{1}{(\gamma)^{1/(q-1)}} \cdot \sup_{y_1, \cdots, y_T \in [0, G]} \sum_{t=1}^{T} \left( \frac{y_t^q}{\sum_{\tau=1}^{t} y_\tau} \right)^{1/(q-1)},$$

where $\mathcal{P}(\mathcal{C})$ denotes the set of probability distributions on $\mathcal{C}$ and where we use $\|x_t\|_2 \leq R \cdot \|x_t\|_{\mathcal{K}} \leq R$ and $\|x_{t+1}\|_2 \leq R \cdot \|x_{t+1}\|_{\mathcal{K}} \leq R$ since $x_t, x_{t+1} \in \mathcal{K}$. The last equality is obtained by independence of $c_t$ and $(c_1, \cdots, c_{t-1})$. The second term is bounded in the proof of Lemma 4.1. As for the first term, we have:

$$\sum_{t=1}^{T} \mathbb{E}[\|c_t\|_2] \cdot (\mathbb{P}[A_t] + \mathbb{P}[B_t])$$

$$\leq 2 \sum_{t=1}^{T} \mathbb{E}[\|c_t\|_2] \cdot \exp(-\frac{\alpha}{16G} \cdot \sum_{\tau=1}^{t-1} \mathbb{E}[\|c_\tau\|_2])$$

$$\leq 2 \sup_{y_1, \cdots, y_T \in [0, G]} \sum_{t=1}^{T} y_t \cdot \exp(-\frac{\alpha}{16G} \cdot \sum_{\tau=1}^{t-1} y_\tau).$$

Observe that, for any $t = 1, \cdots, T$ and for any fixed values $(y_1, \cdots, y_{t-1}, y_{t+1}, \cdots, y_T) \in [0, G]^{T-1}$, the function $y_t \to \sum_{n=1}^{T} y_n \cdot \exp(-\frac{\alpha}{16G} \cdot \sum_{\tau=1}^{n-1} y_\tau)$ is convex on $[0, G]$. Hence, its maximum is attained at an extreme point: either $0$ or $G$. Repeating this process for $t = 1, \cdots, T$, we get:

$$\sum_{t=1}^{T} \mathbb{E}[\|c_t\|_2] \cdot (\mathbb{P}[A_t] + \mathbb{P}[B_t]) \leq 2 \sup_{y_1, \cdots, y_T \in \{0, G\}} \sum_{t=1}^{T} y_t \cdot \exp(-\frac{\alpha}{16G} \cdot \sum_{\tau=1}^{t-1} y_\tau)$$

$$\leq 2G \cdot \sup_{n=0, \cdots, T} \sum_{k=0}^{n} \exp(-\frac{\alpha}{16} \cdot k)$$

$$\leq \frac{2G}{1 - \exp(-\frac{\alpha}{16})} = O(1),$$

which concludes the proof.

## D.4 Derivation of the Lower Bounds

For any given $\alpha \in (0, 1]$, we establish, depending on the curvature of $\mathcal{K}$, lower bounds on regret when the opponent adversarially chooses the hints $(v_1, \cdots, v_T)$ as well as the cost vectors $(c_1, \cdots, c_T) \in \mathcal{C}$. In fact, we establish the regret bounds in the case of a weaker adversary who has to pick a fixed hint $v$ initially and to stick to it throughout the game (i.e. $v_t = v$ for all $t = 1, \cdots, T$). Since this is a weaker notion of adversary, the lower bounds carry over to the more adversarial setting where the adversary is free to pick a different hint at every time period. The minimax regret that can be achieved by an online algorithm in this setting is expressed as:

$$\mathcal{R}_T(\mathcal{C}, \mathcal{K}) = \sup_{v \in B_2} \inf_{x_1 \in \mathcal{K}} \sup_{c_1 \in \mathcal{C}: \, c_1^\mathsf{T} v \geq \alpha \cdot \|c_1\|_2} \inf_{x_T \in \mathcal{K}} \sup_{c_T \in \mathcal{C}: \, c_T^\mathsf{T} v \geq \alpha \cdot \|c_T\|_2} \left[ \sum_{t=1}^T c_t^\mathsf{T} x_t - \inf_{x \in \mathcal{K}} \sum_{t=1}^T c_t^\mathsf{T} x \right],$$

where $B_2$ denotes the unit ball for the $L_2$ norm. Observe that:

$$\mathcal{R}_T(\mathcal{C}, \mathcal{K}) = \sup_{v \in B_2} \Phi(v),$$

where $\Phi(v)$ is the minimax regret that can be achieved by an online algorithm in online linear optimization without hints when the cost vectors $c_t$ all lie in $\mathcal{C} \cap \{c \in \mathbb{R}^d \mid c^\mathsf{T} v \geq \alpha \cdot \|c\|_2\}$ and $x_t \in \mathcal{K}$, i.e:

$$\Phi(v) = \inf_{x_1 \in \mathcal{K}} \sup_{c_1 \in \mathcal{C}: \, c_1^\mathsf{T} v \geq \alpha \cdot \|c_1\|_2} \inf_{x_T \in \mathcal{K}} \sup_{c_T \in \mathcal{C}: \, c_T^\mathsf{T} v \geq \alpha \cdot \|c_T\|_2} \left[ \sum_{t=1}^T c_t^\mathsf{T} x_t - \inf_{x \in \mathcal{K}} \sum_{t=1}^T c_t^\mathsf{T} x \right].$$

Given this characterization, all the lower bounds established in this section are derived by means of a reduction to the lower bounds established in [8] in the standard online linear optimization framework (without hints). Following Flajolet and Jaillet [8], we are first led to identify trivial settings where there is an obvious algorithm that achieves zero regret.

**Definition D.3.** *The "game with hints" is said to be trivial if and only if, for all hints $v \in B_2$, there exists $x(v) \in \mathcal{K}$ such that $c^\mathsf{T} x(v) = \min_{x \in \mathcal{K}} c^\mathsf{T} x$ for all $c \in \mathcal{C}$ that satisfy $c^\mathsf{T} v \geq \alpha \cdot \|c\|_2$.*

If the game with hints is trivial then the optimal strategy is to play $x_t = x(v_t)$ at any time period $t \in \mathbb{N}$ in order to get zero regret. As it turns out, this uniquely identifies trivial games with hints, as we next show.

**Lemma D.4.** *For any $T \in \mathbb{N}$, $\mathcal{R}_T(\mathcal{C}, \mathcal{K}) \geq 0$. Moreover, $\mathcal{R}_T(\mathcal{C}, \mathcal{K}) = 0$ if and only if the game with hints is trivial.*

*Proof.* For the first part, observe that either $\mathcal{C} = \{0\}$ in which case $\mathcal{R}_T(\mathcal{C}, \mathcal{K}) = 0$ or we can find $c \neq 0$ in $\mathcal{C}$ in which case the opponent can pick $v = \frac{c}{\|c\|_2}$ and $c_t = c$ at any time period $t$ which yields:

$$\mathcal{R}_T(\mathcal{C}, \mathcal{K}) \geq \inf_{x_1 \in \mathcal{K}} \cdots \inf_{x_T \in \mathcal{K}} \left[ \sum_{t=1}^T c^\mathsf{T} x_t - \inf_{x \in \mathcal{K}} \sum_{t=1}^T c^\mathsf{T} x \right] = T \cdot \inf_{x \in \mathcal{K}} c^\mathsf{T} x - T \cdot \inf_{x \in \mathcal{K}} c^\mathsf{T} x = 0.$$

For the second part, observe that if the game with hints is trivial, we can play $x_t = x(v_t)$ at any time period $t \in \mathbb{N}$ and the regret incurred is $\sum_{t=1}^T \inf_{x \in \mathcal{K}} c_t^\mathsf{T} x - \inf_{x \in \mathcal{K}} \sum_{t=1}^T c_t^\mathsf{T} x \leq 0$, which in combination with $\mathcal{R}_T(\mathcal{C}, \mathcal{K}) \geq 0$ shows that $\mathcal{R}_T(\mathcal{C}, \mathcal{K}) = 0$. Conversely, suppose that $\mathcal{R}_T(\mathcal{C}, \mathcal{K}) = 0$. Consider any $v \in B_2$. Using Lemma 3 of [8] for online linear optimization when the adversary's decision set is $\mathcal{Z} = \mathcal{C} \cap \{c \in \mathbb{R}^d \mid c^\mathsf{T} v \geq \alpha \cdot \|c\|_2\}$ and the player's decision set is $\mathcal{F} = \mathcal{K}$ (in their notations), we get that $\Phi(v) \geq 0$ and that $\Phi(v) = 0$ if and only if there exists a trivial algorithm for this online linear optimization problem with zero regret, i.e. if and only if there exists $x(v) \in \mathcal{K}$ such that $c^\mathsf{T} x(v) = \min_{x \in \mathcal{K}} c^\mathsf{T} x$ for all $c \in \mathcal{C}$ satisfying $c^\mathsf{T} v \geq \alpha \cdot \|c\|_2$. Since $0 = \mathcal{R}_T(\mathcal{C}, \mathcal{K}) = \sup_{v \in B_2} \Phi(v)$, this implies that the game with hints is trivial. $\square$

Flajolet and Jaillet [8] show that it is essential for $\mathcal{K}$ to be sufficiently curved to get regret bounds better than $\sqrt{T}$ in online linear optimization. We extend this characterization and show that this is true even in the presence of hints. To establish the regret bound, the goal is to find a hint $v^*$ such that

the online linear optimization problem where $c_t \in \mathcal{C} \cap \{c \in \mathbb{R}^d \mid c^\mathsf{T} v^* \geq \alpha \cdot \|c\|_2\}$ and $x_t \in \mathcal{K}$ is non-trivial, in the sense that there is no single point $x^*$ in $\mathcal{K}$ that belongs to $\arg\min_{x \in \mathcal{K}} c^\mathsf{T} x$ uniformly for all $c \in \mathcal{C} \cap \{c \in \mathbb{R}^d \mid c^\mathsf{T} v^* \geq \alpha \cdot \|c\|_2\}$. If we can find such a hint $v^*$, then Theorem 2 of [8] shows that $\Phi(v^*) = \Omega(\sqrt{T})$ and thus $\mathcal{R}_T(\mathcal{C}, \mathcal{K}) = \sup_{v \in B_2} \Phi(v) = \Omega(\sqrt{T})$.

**Theorem D.5.** *Suppose that $\mathcal{K}$ is a polyhedron, then either the game with hints is trivial or* $\mathcal{R}_T(\mathcal{C}, \mathcal{K}) = \Omega(\sqrt{T})$.

*Proof.* Suppose that the game with hints is not trivial. Then, there exists a hint $v^* \in B_2$ such that for all $x \in \mathcal{K}$ there exists $c \in \mathcal{C}$ such that $c^\mathsf{T} v^* \geq \alpha \cdot \|c\|_2$ and $c^\mathsf{T} x > \min_{x \in \mathcal{K}} c^\mathsf{T} x$. As a result, and borrowing the notations of Lemma D.4, $\Phi(v^*) = \Omega(\sqrt{T})$ follows from Theorem 2 of [8] applied to the online linear optimization problem where the adversary's decision set is $\mathcal{Z} = \mathcal{C} \cap \{c \in \mathbb{R}^d \mid c^\mathsf{T} v^* \geq \alpha \cdot \|c\|_2\}$ and the player's decision set is $\mathcal{F} = \mathcal{K}$ (in their notations). Since $\mathcal{R}_T(\mathcal{C}, \mathcal{K}) = \sup_{v \in B_2} \Phi(v)$, this concludes the proof. $\square$

This shows that $o(\sqrt{T})$ regret bounds are not possible when $\mathcal{K}$ is not curved. It is a priori unclear however whether $\log(T)$ is the optimal growth rate when $\mathcal{K}$ is strongly convex. We show that this is indeed the case by means of a reduction to a standard online linear optimization problem where $\mathcal{K}$ is a $L_2$ ball and $0$ does not lie in the convex hull of the adversary's decision set.

**Theorem D.6.** *If $\mathcal{K}$ is a $L_2$ ball, then either the game with hints is trivial or $\mathcal{R}_T(\mathcal{C}, \mathcal{K}) = \Omega(\log(T))$.*

*Proof.* Suppose that the game with hints is not trivial. There exists a hint $v^* \in B_2$ such that for all $x \in \mathcal{K}$, there exists $c \in \mathcal{C}$ such that $c^\mathsf{T} v^* \geq \alpha \cdot \|c\|_2$ and $c^\mathsf{T} x > \min_{x \in \mathcal{K}} c^\mathsf{T} x$. In other words, for all $x \in \mathcal{K}$, we can find a cost vector $c$ that is valid with respect to the hint $v^*$ and such that $x$ is not the optimal solution for $c$. Let $x_1, x_2$ be two non-co-linear points in $\mathcal{K}$ and let $c_1$ as a $c_2$ be the corresponding valid cost vectors. Observe that, necessarily, $c_1$ and $c_2$ are non-co-linear and we have $0 \notin [c_1, c_2]$ since $(\lambda c_1 + (1-\lambda) c_2)^\mathsf{T} v^* \geq \alpha \cdot (\lambda \|c_1\|_2 + (1-\lambda) \|c_2\|_2) \geq \alpha \cdot \min(\|c_1\|_2, \|c_2\|_2) > 0$ for $\lambda \in [0, 1]$. Then $\Phi(v^*) = \Omega(\log(T))$ follows from Theorem 5 of [8] applied to the online linear optimization problem where the adversary's decision set is $\mathcal{Z} = \{c_1, c_2\}$ and the player's decision set is $\mathcal{F} = \mathcal{K}$ (in their notations). Note that the assumptions of Theorem 5 of [8] are satisfied since, as shown above, $0 \notin \mathrm{conv}(\mathcal{Z})$. Since $\mathcal{R}_T(\mathcal{C}, \mathcal{K}) = \sup_{v \in B_2} \Phi(v)$ this concludes the proof. $\square$