[Reviews · NeurIPS 2017]

Reviewer 1



This paper introduces a new setting for online linear optimization where the learner receives a signal at the beginning of each round about the loss function going to be played by adversary. Under strongly convexity assumption on the player’s decision set, the paper shows an improved O(log n) regret bound. The authors also provide nice results on lower bounding the regret under different notions of convexity/uniformity on decision set. The problem being dealt with is interesting and has proper motivation, though previous papers have dealt it with in the past, in one way or another. The presentation of the paper was mostly clear. The claimed contributions are discussed in the light of existing results and the paper does survey related work appropriately. Regarding the quality of the writing, the paper is reasonably well written, the structure and language are good. The paper is technically sound and the proofs seem to be correct as far as I checked. To conclude, the paper is overall well written and well organized. The problem dealt with consists of a new view to the online linear optimization that requires a new type of analysis. The exact setup of the feedback lacks motivation, but overall the idea of an analysis aimed to exploiting side information in online learning is sufficiently interesting and motivated.

Reviewer 2



The paper considers a model of online linear optimization in a convex and compact set K where, at the beginning of each time step t, the learner receives an adversarially chosen vector v_t such that the cosine of the angle between v_t and the actual loss vector at time t is at least alpha > 0. The main result shows that the hint can be used to improve the regret by optimally changing the component parallel to the hint of a prediction x. This corresponds to measuring the linear loss of x using a convex "virtual" loss function whose curvature is related to the curvature of K. In particular, when K is (essentially) a L_2 ball (i.e., modulus of convexity exponent q=2), then the virtual loss is exp-concave. As the virtual loss regret is always smaller than the linear loss regret, this implies a logarithmic regret bound on the original problem. The scenario for q > 2 is not so nice anymore (the problem gets harder because for q large K tends to a L_infinity ball for which hints provably don't help). Indeed, improvements over the T^{1/2} standard regret rate are obtained only when q < 3 and hints are taken from a finite set. Also, the algorithm is completely different from the one for the case q = 2. Instead, here one would like to have a single algorithm covering all q larger or equal than 2, with a regret attaining T^{1/2} as q grows large, and no need for a finite set of hints. The lower bounds are interesting and informative. Overall, an interesting paper with nice results for the case q=2. The virtual loss functions are a good trick that works well. On the other hand, the setting is not really motivated and the results for q > 2 are not that convincing. In "Competing with wild prediction rules" Vovk studied a possibly related problem where the learner competes with a Banach space of functions with curvature q \ge 2 and uses no hints. Perhaps this setting can be reduced to that one.

Reviewer 3



The paper concerns online linear optimization where at each trial, the player, prior to prediction, receives a hint about the loss function. The hint has a form of a unit vector which is weakly correlated with the loss vector (its angle's cosine with loss vector is at least alpha). The paper shows that: - When the set of feasible actions is strongly convex, there exists an algorithm which gets logarithmic regret (in T). The algorithm is obtained by a reduction to the online learning problem with exp-concave losses. The bound is unimprovable in general, as shown in the Lower Bounds section. - When the set of actions is (C,q)-uniformly convex (the modulus of uniform convexity scales as C eps^q), and the hint is constant among trials, a simple follow-the-leader algorithm (FTL) achieves a regret bound which improves upon O(sqrt(T)) when q is between 2 and 3. This is further extended to the case when the hint comes from a finite set of vectors. - When the set of actions is a polyhedron, the worst case regret has the same rate as without hints, i.e. Omega(sqrt(T)), even when the hint is constant among trials. The first result is obtained by a clever substitution of the original loss function by a virtual loss which exploits the hint vector and is shown to be exp-concave. Then, an algorithm is proposed which as subroutine uses any algorithm A_exp for exp-concave online optimization, feeding A_exp with virtual losses, and playing an action for which the original loss is equal to the virtual loss (by exploiting the hint) of A_exp. Since the virtual loss of the adversary is no larger then its original loss, the algorithm's regret is no more than the regret of A_exp. Interestingly, the second result uses FTL, which does not exploit the hint at all (and thus does not even need to know the hint). So I think this result is more about the constraint on the losses -- there is a direction v, along which the loss is positive in each round. This result resembles results from [15] on FTL with curved constraint sets. I think the results from [15] are incomparable to those here, contrary to what discussion in the appendix say. On the one hand, here the authors extended the analysis to uniformly convex sets (while [15] only work with strongly-convex sets). On the other hand, [15] assume that the cumulative loss vector at each trial is separated from the origin, and this is sufficient to get logarithmic regret; whereas here logarithmic regret is achieved under a different assumption that each individual loss is positive along v (with constant alpha). The extension of Theorem 4.2 to arbitrary hints (Appendix C) gives a bound which is exponential in dimension, while for q=2 Section 3 gives a much better algorithm which regret is only linear in the dimension. Do you think the exponential dependence on d for q > 2 is unimprovable? The paper is very well written, and the exposition of the main results is clear. For each result, the authors give an extended discussion, including illustrative examples and figures, which builds up intuition on why the result holds. I checked all the proofs in the main part of the paper and did not find any technical problems. I found the contribution to online learning theory strong and significant. What the paper lacks, however, is a motivating example(s) of practical learning problem(s) in which the considered version of hint would be available. --------------- I have read the rebuttal. The authors clarified comparison to [15], which I asked about in my review.